# Attempts to Create Products with Increased Health-Promoting Potential Starting with Pinot Noir Pomace: Investigations on the Process and Its Methods

**DOI:** 10.3390/foods11141999

**Published:** 2022-07-06

**Authors:** Stephen Lo, Lisa I. Pilkington, David Barker, Bruno Fedrizzi

**Affiliations:** 1School of Chemical Sciences, University of Auckland, 23 Symonds St, Auckland 1010, New Zealand; lisa.pilkington@auckland.ac.nz (L.I.P.); d.barker@auckland.ac.nz (D.B.); b.fedrizzi@auckland.ac.nz (B.F.); 2Centre for Green Chemical Science, School of Chemical Sciences, University of Auckland, 23 Symonds St, Auckland 1010, New Zealand

**Keywords:** Pinot noir pomace, solid–liquid extraction, valorization, flavonoids, fatty acids, derivatization

## Abstract

A process for using grape (Pinot noir) pomace to produce products with improved health-promoting effects was investigated. This process integrated a solid–liquid extraction (SLE) method and a method to acylate the polyphenolics in the extract. This report describes and discusses the methods used, including the rationale and considerations behind them, and the results obtained. The study begins with the work to optimize the SLE method for extracting higher quantities of (+)-catechin, (−)-epicatechin and quercetin by trialing 28 different solvent systems on small-scale samples of Pinot noir pomace. One of these systems was then selected and used for the extraction of the same flavonoids on a large-scale mass of pomace. It was found that significantly fewer quantities of flavonoids were observed. The resultant extract was then subject to a method of derivatization using three different fatty acylating agents. The antiproliferative activities of these products were measured; however, the resulting products did not display activity against the chosen cancer cells. Limitations and improvements to the methods in this process are also discussed.

## 1. Introduction

The International Organisation of Vine and Wine (OIV) recently reported that 248–294 (an average of 266) million hectoliters of wine was produced each year, globally, in between the years of 2016 to 2020 [1]. These figures highlight the scale at which wine is produced to satisfy a large global demand. It is estimated that for every 100 L of wine produced, 18–35 kg of grape pomace (the unwanted grape stems, skins and seeds) exits out as a waste product [2]. There are concerns that this large and accumulating waste mass will become unmanageable and have negative impacts towards the environment [3,4,5,6]. Other than to discard it as landfill, other practices currently employed to manage this waste are to reuse the material either as compost or livestock feed [3,6]. However, employments of these methods to break down the pomace are not without their limitations. The high phenolic content within the material leads to germination problems, and the polymeric compounds (i.e., lignins) poses issues for animal digestion [3]. The environmental concerns have led government bodies to place stringent policies around managing this waste and, in some cases, has resulted in financial penalties on wine producers [7,8]. This has encouraged a need to explore novel strategies in utilizing grape pomace for higher value purposes. Development of these strategies has also been driven by consumer demands for both naturally sourced starting materials and sustainable practices, which include implementing the reusing and recycling of waste products in the production process to achieve a circular economy model [9,10].

Grape pomace contains a variety of valuable compounds, including lignins, aroma precursors, sugars, proteins, unsaturated fatty acids and phenolics [3,4]. These compounds have properties that can be exploited to provide benefits to consumers and are attractive for various commercial applications. Phenolics (which includes polyphenols) in grape pomace, for example, comprise of a diverse set of compounds that are further classified into groups (such as phenolic acids, flavonoids and stilbenes) based on their core chemical structures. Polyphenols are well known for their biological activity and have demonstrated their ability to exert antioxidant, antiproliferative, anti-inflammatory, antimicrobial and neuroprotective effects [3,4,11,12]. It has been proposed that the recovery of valuable compounds from grape pomace could be further processed and applied for purposes such as food fortification [13], food processing [9,14], food preservation [15], improving wine quality [16], antibacterial materials [17] and cosmetic use [18]. Additionally, given their bioactive properties, it has been suggested that polyphenolic compounds could be sourced as starting materials and converted into products that directly deliver health-promoting effects [4,19]. In the literature, there are a number of examples in which grape pomace extracts, which have higher or more concentrated phenolic contents, were found to exhibit antioxidant, antimicrobial, anti-inflammatory and antiplatelet activities [20,21,22], as well as promising antiproliferative or cytotoxic activities against cancer cells [23,24]. These studies were validated based on chemical, biochemical and in vitro studies and are promising indications for the health-promoting potential of the pomace. There are, however, fewer examples of in vivo studies that demonstrate the health-promoting effects of grape pomace products in animals or humans. As one example, Olivero-David et al. demonstrated that their grape by-product extract was able to the lower cholesterol levels in Wistar rats, which has implications towards hypercholesterolemia [25]. The scarcity of such studies is due to the fact that bioactive polyphenols have poor oral bioavailability, and this severely limits their effects in live models [26,27]. One major contributing factor to their low bioavailability is that these compounds are highly susceptible to metabolism by the enzymes they encounter as they pass through compartments such as the intestinal cells and the liver before reaching the systemic circulation [28,29]. These enzymatic activities convert these compounds into metabolite products that are inactive and more easily eliminated from the body. This is a large barrier that needs to be overcome. The ability to increase their bioavailability enables higher amounts of the active compounds to enter into the systemic circulation, be transported to the cells and tissues of interest and be able to exert the desired health-promoting activities. Further, if these compounds can be structurally manipulated to have increased activity (i.e., increased binding affinity to their biological targets), then this reduces the amounts of active compounds required around the active sites to exert these health effects. Therefore, the strategy for developing health-promoting products from grape pomace must thoroughly consider a process that can both effectively extract the bioactive polyphenolic compounds from the waste material and make the appropriate adjustments to these compounds that would enhance the properties needed for health promotion, including increased bioavailability and bioactivity.

The aim of the study was to develop a process whereby bioactive polyphenolics could be sourced from grape pomace, used as starting materials and modified into products with improved health-promoting properties. This work drew inspiration from previously reported studies conducted by our research group, which presented findings on the use of a simple method for extracting polyphenolics from grape pomace [30] and the structural modifications to flavonoids that led to improved bioactivity [31,32]. It was proposed that the integration of these methods could be a feasible process to achieve the study aim. Unfortunately, in following this proposed process, the expected outcome was not achieved. Despite that, the investigations of these methods did generate results that are worth presenting. This report aims to describe and discuss these methods, including the rationale and considerations underlying the decisions to use and adapt them, and the results. Following that, limitations to these methods are explored and improvements are suggested.

## 2. Materials and Methods

### 2.1. Chemicals and Reagents

Commercial standards include (+)-catechin hydrate (≥98%), (−)-epicatechin (≥90%), rutin hydrate (≥95%) and quercetin (≥95%) were purchased from Sigma-Aldrich (Auckland, New Zealand). Acetonitrile (MeCN, HPLC grade), L-(+)-tartaric acid (reagent grade), acetone (analytical grade) and ethanol (analytical grade) were purchased from ECP (Auckland, New Zealand). Glacial acetic acid (AcOH, HPLC grade) was purchased from ThermoFisher Scientific (Auckland, New Zealand). Ultrapure water (H_2_O) had resistivity of 18.2 MΩ cm and was obtained from Sartorius Arium^®^ Pro system. Pinot noir grape pomace (seeds, skins and stems) was industrially sourced from a New Zealand vineyard in 2015. Acylating agents (octanoyl chloride, lauroyl chloride and palmitoyl chloride) were purchased from Sigma-Aldrich (Auckland, New Zealand).

### 2.2. Preparation of Model Red Wine and Solvent Systems

Model red wine was prepared by dissolving tartaric acid (5 g) into 13.5% EtOH in ultrapure water (1 L). The pH of this solution was adjusted to 3.5 with NaOH (10 M and 1 M).

A subset of 28 systems were selected (from those used in the literature) and trialed in extraction on small-scale mass Pinot noir pomace samples (see Section 3.1) [30]. The ratios of the solvents (acetone, H_2_O and EtOH) that made up the selected systems (50 mL) are found in Table 1.

### 2.3. HPLC Analysis

All standards and samples were analyzed using the Agilent 1290 Infinity Liquid Chromatography (Santa Clara, CA, USA), fitted with a quaternary pump and coupled to a UV-Vis DAD. DAD wavelengths were set to 280 nm for (+)-catechin hydrate and (−)-epicatechin; 350 nm for rutin hydrate; and 365 nm for quercetin. The eluents are 100% H_2_O (solvent A), 5% AcOH in H_2_O (solvent B) and 100% MeCN (solvent C). The HPLC method used was reported in the literature (Appendix A) [33]. The standards and samples were eluted through the Phenomenex Luna C-18 Column (250 mm × 4.6 mm, 5 µm particle size, Phenomenex, Torrance, CA, USA) at a flow rate of 0.8 mL/min at 25 °C.

Retention times (tr) for (+)-catechin hydrate, (−)-epicatechin, rutin hydrate and quercetin were 38.0 min, 54.5 min, 76.1 min and 94.6 min, respectively. External calibration standard curves for (+)-catechin hydrate, (−)-epicatechin and rutin hydrate were generated, with each curve consisting of six calibration points that were measured in triplicate. Each curve displays excellent linearity with a correlation coefficient (r^2^) of >0.99. The limit of detection and limit of quantitation for each flavonoid per kg pomace are 13.2 mg and 43.9 mg for **1**; 1.9 mg and 6.6 mg for **2**; and 2.8 mg of rutin hydrate and 9.4 mg (of rutin) for **3**.

### 2.4. Materials and Solid–Liquid Extraction Protocol

The solid–liquid extraction (SLE) protocol used in this study was similar to that described in the literature with some adaptations (discussed in Section 3.1 and Section 3.2) [30]. Pinot noir pomace was homogenized with a kitchen blender. For the small-scale extraction study, the pomace samples (10 g) were then immersed in the solvents (50 mL) and stirred at r.t. for one hour in an open vessel. The solvent was decanted and then centrifuged at 6000 rpm for 10 min. The supernatant (5 mL) was taken, and the solvent removed in vacuo. The resulting extract was dissolved in model red wine (2 mL). This solution was passed through a Phenomenex regenerated cellulose membrane syringe filter (0.45 μm pore size, 15 mm diameter, from CA, USA) and stored at −20 °C until HPLC analysis. The quantities of each flavonoid in each of the samples were analyzed with HPLC according to the method described in Section 2.3. Extractions of the pomace samples with each solvent system were conducted in triplicate.

For the large-scale extraction, Pinot noir pomace (2 kg) was separated into batches (200 g). Each batch was immersed in 40:30:30 acetone:H_2_O:EtOH solvent (1 L) and manually stirred for 1 h. After extraction, the solid–solvent mixture was poured over a kitchen sieve to filter out the solid mass and to collect only the phenolic-enriched solvent, which was then centrifuged. For quantification, the supernatant (10 mL) from each batch was taken and combined together (total of 100 mL). The solvent was removed in vacuo. Model red wine (6 mL) was then added to the extract and separated into 3 equal portions (2 mL), which were passed through the Phenomenex syringe filter (0.45 μm pore size, 15 mm diameter, from CA, USA). The quantities of each flavonoid in each of the samples were analyzed with HPLC. The rest of the phenolic-rich solvent were combined together, and the solvent was removed in vacuo to give a resulting extract that would be used for derivatization. Various stages of this process can be seen in Figure 1.

### 2.5. Derivatization of the Flavonoid-Enriched Extract and Infrared Analysis

The starting material extract (1 g) was dissolved in dimethyl formamide (DMF, 15 mL) solvent. Triethylamine was added to this solution and then the acylating agents (octanoyl, lauroyl and palmitoyl chlorides). The reaction was stirred for 24 h and then was quenched with NaHCO_3_. The mixture was then filtered and washed with excess water. The resulting products were collected and dried to give red, solid products. The infrared (FT-IR) spectra of non-derivatized and derivatized products were obtained using Perkin-Elmer Spectrum 100 FT-IR spectrometer (Santa Clara, CA, USA).

### 2.6. Antiproliferative Activity Procedures

The antiproliferative studies on HCT116 and MDA-MB-231 cell lines were conducted using the [^3^H]-thymidine incorporation assay method, as previously reported by [34]. Essentially, the method was conducted by seeding 3000 cells in each well of the 96-well plates with a solution of non-derivatized and derivatized extracts in dimethyl sulfoxide (DMSO) for 3 days. Then, [^3^H]-thymidine is added to the cells, and they are incubated for 6 h. The cells were then counted using a Trilux/Betaplate counter, which shows the percentage of the cells with [^3^H]-thymidine that are incorporated into the DNA helix. This was conducted in duplicate. The antiproliferative activity was determined as cell growth percentages relative to the 100% growth in the control (non-treated).

### 2.7. Statistical Analysis

Ternary plots were used to identify and depict regions of high and low concentrations of compounds extracted across the investigated solvent ratios. R (version 3.6.0) was used to conduct the analysis using the “ggtern” R package [35,36]. To create the plots, interpolation was carried out using a multivariate linear regression with a fitted polynomial expression that was chosen on the basis of best fit and accurate representation of the measured data.

Statistical analyses of the antiproliferative testing were conducted using one sample, a one-tail *t*-test to determine whether the mean proliferation value of each product was lower than that of the control (100%) and a one-way ANOVA to determine whether there were any differences between these values.

## 3. Results and Discussion

### 3.1. Optimization of the Solid–Liquid Extraction Method (Small-Scale Extraction Studies)

This study began with an investigation into an appropriate method for extracting polyphenolics from grape pomace. Many considerations were made to develop this part of the process. These included the variety of grape pomace that would be used, the specific polyphenolic compounds to target, the selection of an appropriate extraction technique and how the extraction protocol could be optimized.

In consideration of the pomace, it was decided to use a grape variety that was of significance to New Zealand (NZ). Sauvignon blanc has been recognized as the most significant grape to NZ, as its production mass is the highest of all grapes (with an annual production mass of around 10-fold higher than the grape variety that follows behind it) [37,38]. Our research group had already investigated the use of this grape pomace, including the extraction of valuable compounds (aroma precursors and phenolics) from it and processes to valorize them [17,30]. Alternatively, Pinot noir is another grape type that is of significance to NZ, with a production mass (average of 30,000 tonnes produced each year between the years from 2012 to 2021) often following behind Sauvignon blanc as the second most produced grape type in the country [37,38]. For this reason, Pinot noir pomace was selected for use in this study.

It was mentioned earlier that grape pomace contains a wide range of polyphenolic compounds. Among them, the flavonoid compounds are one prominent class. In line with polyphenolics, flavonoids also possess a wide range of biological activities, including antioxidant and antiproliferative activity, as well as potential for anti-inflammatory, anti-thrombotic, cardio-protective and neuro-protective effects [19,39,40]. Therefore, it was considered that the extraction strategy should be tailored to selectively target the flavonoids that are present in the pomace. Raising the profile of these compounds also serves to assist the next method in the process, which was chosen based on knowledge about the lipophilic flavonoid derivatives and their improved bioactivity (see Section 3.3). Upon review of the literature that have reported the extraction of polyphenolics (particularly focusing on SLE extraction techniques) from grape pomace, it was noted that (+)-catechin (**1**), (−)-epicatechin (**2**) and quercetin (**3**), are the most frequently cited flavonoids (Figure 2 and Table 2) [20,30,41,42,43,44,45,46]. Therefore, it was decided that the extraction method would be developed to obtain as much of these three flavonoids as possible.

In the literature, a wide variety of extraction techniques have been presented for extracting phenolics from plant-based materials. Examples include conventional SLE [30,41], SLE assisted with ultrasound [47,48], SLE assisted with microwave [49,50], and supercritical fluid extraction [51,52]. Each technique has advantages (or disadvantages) based on factors such as cost, ease of use, efficiency of extraction, selectivity of phenolics, requirement for specialized equipment and being environmentally friendly. Since this study intended to be an initial proof of concept, the primary criteria for selecting the extraction technique were simple to conduct, relatively low cost and reduced reliance on sophisticated and specialized equipment. The conventional SLE satisfies these criteria, and, therefore, this technique was chosen. In the literature, this technique has been employed on different types of grape pomace, and selected examples of these works and the quantities of flavonoids **1**, **2** and **3** extracted are presented in Table 2.

The SLE protocol selected for this study was based on one that had previously been implemented by our research group [30]. In that study, the SLE technique was investigated for its ability to extract a wide range of aroma precursors and phenolic compounds from Sauvignon blanc pomace. The study trialed the protocol with a series of 66 different solvent systems (made up of the three different polar solvents—acetone, H_2_O and EtOH) to identify the optimal systems for extracting each compound of interest. From those findings, the optimal systems proposed for flavonoids **1**, **2** and **3** were of relevance to the purpose of the present study (Table 2 and Figure 3). However, before proceeding to apply the SLE protocol with the exact same solvent systems to the Pinot noir pomace, it was considered that those findings were specific to Sauvignon blanc pomace and that differences in the pomace matrix between the two grape types could result in differences in the extraction outcomes. In essence, what is optimal for one pomace is not necessarily the same for another. Therefore, it was necessary to conduct another trial to independently ascertain which solvent systems would be optimal for extracting these three flavonoids from Pinot noir pomace. Rather than conducting the same trial with all 66 solvent systems, a more focused approach was taken by narrowing down the number of systems in the trial. Based on an analysis of the ternary diagrams for each of the three flavonoids (presented from the report by Jelley et al.), a subset of 28 solvent systems were considered as having generally good ability to extract those desired flavonoids (see Table 1 and depicted as the red triangle on Figure 3) [30]. Therefore, it was decided that these solvent systems would be selected for the trial. The trial was conducted on a small mass (10 g) of Pinot noir pomace. It is envisaged that, by determining the quantities of flavonoids extracted from each system, this would then help in the identification of the few systems that are optimal. From these, one would then be selected and taken forward for use in the large-scale pomace mass extraction study (see Section 3.2).

The small-scale extraction trial was conducted according to Section 2.4. At the end of each extraction (and after the solvents had been removed in vacuo), a pomace extract resulted, all of which were viscous residues that were red in color. Prior to HPLC analysis, this extract was dissolved in a small volume of model red wine. The reason for this was the need to dissolve all of the contents in the extract in a solvent with low volatility to minimize solvent loss through evaporation. Model red wine was an ideal choice as it was low in volatility and, proposedly, its resemblance to actual red wine enabled dissolution of all the contents in the extract. The dissolved extracts were then analyzed with HPLC to determine the quantities of the three flavonoids. The quantities of these flavonoids extracted from each individual solvent system are found in Appendix A. From these results, it was found that the optimal systems for extracting **1**, **2** and **3** were the 80:20:0, 40:40:20 and 40:50:10 systems, respectively (Table 3).

To develop some perspective on the performance of this method, these flavonoid quantities were compared (by determining the relative and absolute differences) to those reported in the selected literature presented in Table 2. This provided an indication as to whether the SLE method used in the present study was more, less or the same in effectiveness for extracting these compounds compared to the method employed by others. It should be noted that there are significant differences between the SLE protocol used in the selected literature and that of the present study. These differences include factors such as grape type that was used, the pomace component (i.e., whole pomace, only skins, only seeds), preparation of the pomace (i.e., lyophilized, blended, grounded), solvent systems (i.e., solvent type, mono or multi, different ratios) and the conditions (i.e., temperature and length of extraction time). The limitation of comparing results that have been obtained from very different methods is the inability to deduce the extent to which each factor would have contributed to the quantitative differences. In comparison, the quantities of **1**, **2** and **3** obtained from the Pinot noir pomace were 2.7-fold lower (by 292.2 mg/kg pomace), 1.8-fold higher (118.5 mg/kg) and 2.8-fold higher (55.1 mg/kg), respectively, compared to those obtained from the Sauvignon blanc pomace [30]; the quantities of **1** and **2** were 1.3-fold lower (52.6 mg/kg) and 2.0-fold higher (135 mg/kg), respectively, compared to those obtained from the Weisser Riesling pomace skins [45]. These differences were not large and suggest that the present method was slightly less effective for **1**, slightly more effective for **2** and slightly more effective for **3**, compared to that of the two studies. In contrast, the differences in the quantities of the flavonoids were much more pronounced compared to those obtained from the pomace seeds. For example, quantities of **1** and **2** in the present study were 4.5-fold lower (616.1 mg/kg) and 2.5-fold lower (404.9 mg/kg), respectively, to those obtained from the Weisser Riesling pomace seeds [45]; the quantities of **1** and **3** were 8- to 16-fold lower (1299 to 1409 mg/kg) to those obtained from the Pinot noir seeds (extracted using both MeOH and EtOH solvent) [46]. The increased effectiveness of those methods for these flavonoids may be attributed to the fact that seeds in general carry higher amounts of these flavonoids. This notion is supported by comparing the results within the report by Kammerer et al., which showed that the contents of **1** and **2**, extracted from the pomace seeds, were much higher than those obtained from the pomace skins [45].

For a more general understanding on the solvent systems’ effectiveness for extracting these flavonoids, the data (average quantities of the flavonoids extracted) were statistically analyzed and visually interpreted as model ternary diagrams (Figure 4b–d). It should be noted that the data displayed in these diagrams fall within a range of solvent systems (depicted as a green triangle in Figure 4a) that differ from the experimental range (depicted as a red triangle in Figure 4a). Therefore, the limitations of these diagrams are that the data for solvent systems with <10% EtOH are omitted and that the data generated for solvent systems with <20% water were extrapolated. From analyses of each individual diagram, it was observed that the extraction of **1** from the pomace was better achieved with systems containing higher amounts of acetone (50–80%) combined with low amounts of H_2_O (10–30%) and EtOH (10–30%). The extraction of **2** was better achieved with systems containing moderate amounts of acetone (40–60%) combined with low amounts of H_2_O (10–30%) and low-to-moderate amounts of EtOH (20–40%). The extraction of **3** was better achieved with systems containing moderate amounts of acetone (40–50%) combined with moderate amounts of H_2_O (30–50%) and moderate amounts of EtOH (30–40%). From an analysis of all three ternary diagrams together, it was deduced that these six systems–60:30:10, 50:40:10, 50:30:20 40:50:10, 40:40:20 and 40:30:30 (depicted as a blue triangle in Figure 4a)—would be optimal for the overall extraction of all three flavonoids. This is supported by looking at the ranking of these solvent systems (out of all 28) for each compound (Table 4). Five of these systems were found to be within the top five in their ability to extract at least one of the three flavonoids, and the other system ranked just above the middle for all three flavonoids. A process was then taken to narrow down further the most suitable solvent system that would be used in the SLE method for extraction of flavonoids from a large-scale mass of pomace. This process was executed by choosing the top few systems based on the sum of ranks and then validating these systems according to the ternary diagrams. By looking at the sum of ranks, the top three were 40:40:20 (sum = 12), 50:30:20 (14) and 40:50:10 (17). These systems ranked within the top five for extracting two flavonoids and were ranked as being moderately effective for the other flavonoid. The next best system was 40:30:30 (24), which was relatively good for all three flavonoids. In comparing these systems on the ternary diagrams, it was seen that 40:50:10 was the least effective for **1** and **2** and that 50:30:10 was the most effective for **2** but least effective for **3**. The 40:30:30 and 40:40:20 systems were similar to each other in that they were both effective for all three flavonoids. It was reasoned that, out of the two, the 40:30:30 system had a better balance of all three solvent systems. For this reason, it was decided that this system would be used for the subsequent SLE of a large-scale mass (2 kg) of Pinot noir pomace.

### 3.2. Large-Scale Solid–Liquid Extraction of Pinot noir Pomace

After identifying an optimal solvent system from the small-scale trials, the next part of this study was to implement the SLE protocol on a large-scale mass of the Pinot noir pomace. It was expected that this would deliver a larger mass of the extract and, therefore, higher total amounts of the three flavonoids. In this part of the study, the pomace mass was scaled up (from 10 g) to 2 kg, which was thought of as both a mass that was manageable and also appropriate for a proof of concept study. It was intended to keep all procedures similar to those of the small-scale extraction; however, some adjustments were needed. To remain consistent with the pomace’s mass-to-solvent volume ratio, the 2 kg mass would have needed to be immersed into 10 L of solvent. This was considered impractical for a few reasons. Firstly, without bespoke equipment, the handling of such a large volume of volatile solvent (acetone and MeOH) posed safety issues; secondly, this would have required a specialized stirring apparatus to consistently move a large body of pomace around the large volume of solvent for the duration of the extraction time. Thirdly, the transfer of solvent from one vessel to another would have resulted in unavoidable and non-ideal spillages and, thus, the loss of both solvent and content. To mitigate these issues, the 2 kg pomace mass was divided into 10 batches of 200 g of pomace, in which each batch was immersed into 1 L of solvent. Another adjustment was made to the way in which the pomace was moved around the solvent. Consistent motion of the pomace around the solvent was necessary to ensure maximum contact between each pomace particle and the solvent. However, in the absence of an appropriate stirring apparatus (possessing sufficient power to move the thick pomace around the liquid), the stirring was conducted manually. Stirring of some capacity is a better alternative to non-stirring; however, this manual stirring, which is further addressed in Section 3.5, was a major limitation of this method. At the end of the extraction, the resultant extracts had the same appearance (red-colored viscous residue) as that which was observed from the small-scale extraction. The extracts obtained from each batch were combined, and the quantities of the flavonoids in the combined extract were analyzed.

It was found that the quantities of the flavonoids obtained from the large-scale extraction were lower, each to varying degrees, compared to those obtained from the small-scale extraction using the same solvent system (Table 5). The amount of **1** and **2** was significantly reduced by 10 times and 240 times, respectively, and with absolute differences of 150.7 mg/kg and 239.0 mg/kg of pomace, respectively. As for the amount of **3**, this difference was less pronounced and had only reduced by half, with an absolute difference of 25.2 mg/kg of pomace. It is speculated that one of the reasons for the less pronounced difference observed between the quantities of **3**, compared to those of **1** and **2,** is the fact that the 40:30:30 solvent system comprises moderate amounts of all three solvents. As discussed in Section 3.1, the systems with moderate amounts of the three solvents were found generally to be more suitable for extracting this flavonoid. There are some limitations to this extraction method that would have contributed to the reduced amounts of all three flavonoids, which are further discussed in Section 3.5.

### 3.3. Fatty Acyl Derivatization of the Extract

Despite the smaller than expected amounts of flavonoids obtained from the large-scale extraction method, it was still decided to take the pomace extract forward to the next method of the process. The lack of flavonoids in the extract required the original study concept (of developing an extract enriched with derivatized flavonoids) to be reframed. It was considered that these small amounts of flavonoids were present in a mixture with other phenolic compounds. This study did not implement methods to identify and measure quantities of other bioactive compounds in the extract, but, according to literature, it is suggested that there would be other flavonoids (i.e., anthocyanins and anthocyanidins), stilbenes (i.e., resveratrol) and phenolic acids (i.e., caffeic acid, *p*-coumaric acid) available [3,4]. Thus, it was proposed that subjecting the extract to the derivatization method would lead to structural modifications in the bioactive phenolic compounds, as a collective, and that this would raise the health-promoting properties in the final product compared to those of the original starting extract.

The rationale for derivatizing these compounds is based on the reported evidence regarding structurally modified phenolic compounds that have enhanced biological activities [53,54] and properties associated with bioavailability [55,56,57]. Additionally, our group had also reported studies on fatty acyl modifications to flavonoid structures and their effects on bioactivity. In these studies, it was found that by selectively acylating specific hydroxy positions of both luteolin [32] and quercetin [31], these products displayed improved antiproliferative activity against both HCT116 and MDA-MB-231 cancer cells. It was also found that the radical scavenging activity did not change, which was a good indication that the modifications did not reduce their antioxidant activities. Although these studies did not investigate whether they had properties related to good bioavailability, there are examples in the literature in which an increase in the lipophilicity of phenolic compounds could improve their absorption and metabolic stability [55,56,57]. These findings led to the proposal that acylation modifications to the polyphenolics in the extract, especially the flavonoids, could also result in an enhanced overall health-promoting potential. The feasibility of this proposal is further supported by the success of a similar work, recently reported by Lei et al., who produced fatty acyl derivatives of polyphenolics in grape seed extract and demonstrated their radical scavenging activity and antiproliferative effects against HepG2 cells [58].

The were some considerations around the method for derivatizing the compounds in the extract. Since our reported studies identified three acylated flavonoids (which are octanoyl, lauroyl and palmitoyl derivatives of both luteolin or quercetin) as having better antiproliferative activity, it was decided that the corresponding acylating agents would be used [31,32]. The differences between previous studies and the present study that would affect the acylation approach were also considered. Where previous works were able to utilize a multi-step synthetic approach, starting with a single flavonoid and developing reaction steps to selectively acylate only one hydroxy position on the flavonoid, the same synthetic approach would not be achievable on the pomace extract. The extract comprises a mixture of many different polyphenolic compounds with many hydroxy sites, which are all susceptible to acylation. It was therefore accepted that the acylation approach in this study would be much less targeted and more random (occurring on many compounds and at multiple hydroxy sites) and would give a mixture of different fatty acyl products. 

The method of derivatization was conducted by dissolving 1 g samples of the extract in DMF. This solvent was chosen for its moderate polarity (with a relative polarity of 0.386 [59]), and, therefore, it has the ability to dissolve both the polar polyphenolic content in the extract as well as the hydrophobic acylating agents. This was important as it increases the chances of these compounds contacting and colliding, thereby facilitating the reaction. The reaction was initiated by adding the organic base, triethylamine, to remove hydroxy protons from the phenolics. The acylating agents (either octanoyl chloride, lauroyl chloride or palmitoyl chloride) were then added to acylate these deprotonated hydroxy sites. At the end of the reaction, the mixture was then quenched with NaHCO_3_, filtered and washed excessively with water to remove any remaining salt byproducts. Interestingly, the products that resulted from this procedure were red solids, which is different to those of the starting material.

The non-derivatized and the fatty-acyl-derivatized extract products were characterized by infrared (IR) spectroscopy (Appendix A). From the IR spectrum of the non-derivatized extract, there is a strong broad signal observed at 3275 cm^−1^, representing the hydroxy groups. In this same region in the IR spectra of the derivatized extract products, the intensity of this signal significantly reduced, which indicates that some of the hydroxy groups of the polyphenolics were masked, likely due to the fatty acyl derivatization.

### 3.4. Antiproliferative Activity Studies of the Products

HCT116 and MDA-MB-231 were chosen for study as they are cancer cell lines representative of two very significant cancer types: colon (or colorectal) cancer and breast cancer, respectively. In 2020, colorectal cancer was the third most prevalent cancer for all sexes, and breast cancer was the most prevalent cancer in females [60]. There is a clear need to develop agents that can target the cell lines contributing to these cancer types. The production of lipophilic flavonoids and their improved activity against these two cell lines (compared to that of their parent compounds) have previously been exemplified in the literature. For example, Omonga et al. demonstrated that *O*-alkyl derivatives of chrysin increased activity against HCT116; whilst Nair et al. demonstrated fatty esters of phloridzin increased the inhibitory effects against MDA-MB-231 [61,62]. Further, our group also found that lauroyl, octanoyl and palmitoyl derivatives of quercetin and luteolin had the greatest improvements against these two cell lines. As discussed in Section 3.3, this provided the rationale for derivatizing the polyphenolics in the extracts with these same acyl groups, in hopes that this would raise the extracts’ ability to inhibit the growth of these two cell lines.

The ^3^[H] thymidine incorporation assay was used to study the antiproliferative activity of these extracts against the two cell lines and was conducted according to Section 2.6. Unfortunately, when preparing the extract samples, it was found these products lacked the important solubility in the DMSO solvent, and thus, this was a significant limitation to the results obtained. This lack of solubility would have reduced the amounts of bioactive products available in solution. This unexpected problem was a limitation to this study. Despite that, the procedure was carried out, and the proliferation rates of the cell lines treated with the products were obtained (Table 6). From statistical analyses of these values, it was found that the proliferation rates of these products (both non-derivatized and non-derivatized) were not significantly lower (*p*-value > 0.05, one sample one-tail *t*-test) compared to that of the control (100%). Therefore, there is no evidence that these products limited the growth of these cells. There was also no evidence of a statistical difference (*p*-value > 0.05, one-factor ANOVA) between the mean values of each of the products. Although these results were not promising, they do not necessarily imply a lack of the desired bioactivity. Improvements to this methodology are explored in the following section.

### 3.5. Limitations to the Methods and Suggested Improvements

Overall, this process was not successful in achieving the study aim. However, exploring some of the limitations to the methods provides insights into improvements that could be made. This discussion remains within the scope of using SLE to extract polyphenolics (particularly the flavonoids) from Pinot noir pomace and the lipophilic derivatization of these polyphenolics. Within this scope, there are three areas of the process that are explored. These are the pomace’s preparation, SLE method for large-scale Pinot noir pomace and the method for acylating polyphenolics from the extract.

One limitation was that the pomace that was used was a wet material and contained significant amounts of water, present on the surface of the pomace particles and inside the pomace particles. The implication of using wet pomace is that when it is immersed into the solvent systems, the water passes into the surrounding solvent and raises the water profile in the system. It was discussed in Section 3.1 that flavonoids are generally best extracted with low water solvent systems or that higher water systems are less effective for these compounds. Therefore, an increase in the proportion of water could reduce the ability for the flavonoids to be extracted. To mitigate this limitation, the pomace material could be lyophilized to eliminate water from the materials before extraction.

The pomace material was prepared by blending in attempt to increase homogeneity. This also reduced the pomace particle sizes and, therefore, increased the surface-area-to-volume ratio. However, the resultant pomace particle size after blending may not have been small enough. The importance of lowering this ratio as much as possible is that it enables more solvent to come into contact with more parts of the pomace, thereby maximizing the extraction ability. It is suggested that a more superior preparation process would be to mechanically grind the pomace down into a finer powder. This step would require the pomace material to be dry and, therefore, should be completed after conduct from the first suggestion.

The method used immersed the pomace in the solvent for 1 h, which may not have been long enough to enable total extraction. If the extraction time was extended to at least 6 h, and up to 12 h or 24 h, this could have significantly maximized the extraction. Either alternatively or additionally, the extraction could also be maximized by repeating the extraction, where the pomace is immersed again into freshly prepared solvent. Furthermore, it was questioned whether the conduct of extraction at room temperature was sufficient. It is proposed that raising the temperature (i.e., from 20–25 °C) could improve solubility of the compounds in solvent, and, therefore, it would be worthwhile to investigate the effects of temperature. However, with higher temperatures, the mitigation of solvent (acetone and EtOH) evaporation must be carefully considered. 

The choice to manually stir the pomace in solvent in the large-scale extraction was a great limitation to the method. The motion was inconsistent, and it was reasoned that the lack of even contact between the pomace particles and the solvent was a major contributor to the lower amounts of flavonoids extracted from the large pomace mass. A suggested improvement is to implement an overhead mechanical stirrer (with sufficient power) to produce a continuous and consistent motion for moving the pomace around the solvent.

From the small-scale extraction investigation, six solvent systems were identified as ideal (see Section 3.1). However, only one system was taken forward and investigated on the large-scale pomace mass. Another trial, using all six systems on the large-scale pomace, could be conducted to validate which system is the best system for extracting the desired flavonoids.

This study placed high emphasis on extraction and analysis of the flavonoids **1**, **2** and **3**. The original concept operated under the premise that these flavonoids and their lipophilic derivatives would be able to contribute to enhancing the health-promoting properties of grape pomace extract. One significant improvement would be to expand the investigation to understand the profile of other bioactive phenolic compounds that would typically be found in grape pomace. This would further increase the overall understanding of the components in the extract that could also contribute to the desired health effects.

Returning the focus back onto flavonoids, these compounds existed in a mixture and among a number of other polar compounds in the extract. The limitation of this is that the subsequent derivatization could not be selective of only those compounds. It would be more ideal if these flavonoids (either as a collective or individually) could be isolated and then be subjected to the derivatization step. Alternatively, the process could be followed through to the end, and then the acylated flavonoid derivatives in the resulting product could be purified from the mixture. One possible way of doing this is to use preparative HPLC, which is an effective technique for purifying compounds. However, the tradeoff to this is the high costs of both the equipment and its operation.

Although the reduced intensities observed on the IR spectra are promising indications that the phenolic compounds in the extract may have been masked by acylation, the limitation to this characterization technique is that it remains indicative and does not specify which compounds are present. The use of HPLC–mass spectrometry and (proton and carbon) nuclear magnetic resonance (NMR) are suggested as further characterization techniques that could provide more detailed information on these compounds. However, implementation of these techniques does require some further preparative work. For example, HPLC–mass spectrometry would require an appropriate method that enables good separation of the lipophilic compounds, and NMR would require a degree of purity in order to resolve the signals corresponding to the (proton and carbon) atoms for each compound.

One major limitation to the final derivatized products’ antiproliferative activities was their lack of solubility in the solvent that was required to perform the assay. It is hard to ascertain what exactly caused the lack of solubility. It would be worth conducting a quick assessment on the solubility of this product with a range of other solvents that are appropriate for this test. Another lengthier, but more ideal, improvement is to implement the already mentioned suggestion of further purifying the compounds within the product. This would enable individual (or fractions of only a few) compounds to be prepared separately, and, for those compounds that can be solubilized, these can be taken forward for testing. In turn, this would also help in the identification of compounds (or fractions of compounds) that display activity against the chosen cell lines.

The above suggestions were made with the aim of increasing the prospect of obtaining a final product with improved health-promoting outcomes. However, each additional step introduced to the process will unavoidably increase the cost, complexity, and time. These factors have opposing effects on the commercial viability of the strategy. Therefore, before the adoption of any improvements to the process, the tradeoffs, such as whether the additional benefits will outweigh the drawbacks, and how these drawbacks can be attenuated must be carefully considered.

## 4. Conclusions

This study investigated a process that uses the SLE method on Pinot noir pomace, which was optimized to extract as many of the three flavonoids (**1**, **2** and **3**) as possible and a method that derivatizes the polyphenolics in the extract, converting them into fatty acyl derivatives. It was hoped that this process would result in products that could display some improved health-promoting properties. Unfortunately, this process did not achieve the expected outcome; however, the findings from these investigations remained interesting. In the small-scale extraction study, the trial of 28 selected solvent systems, led to identification of the most effective solvent system for each individual flavonoid and the identification of 6 systems that were ideal for extracting all three flavonoids. Implementing one of these solvent systems for the SLE of large-scale mass of pomace, interestingly, led to significantly reduced quantities of the flavonoids compared to those observed in the small-scale study. Despite that, this extract was taken into the next derivatization step. The IR spectra of these products was used to indicate the success of this derivatization. The subjection of these products to antiproliferative testing did not display improved results. The exploration of the limitations to these methods and suggested improvements may serve as foundations for future work with possibilities to increase success.

## Figures and Tables

**Figure 1 foods-11-01999-f001:**
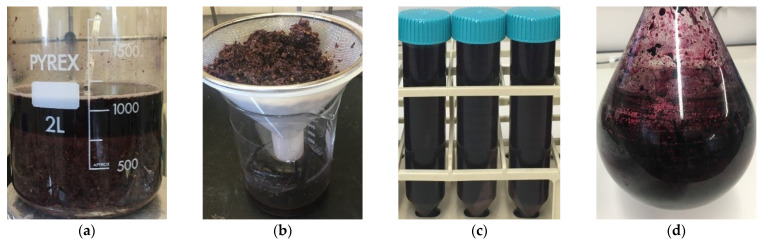
Large-scale Pinot noir extraction procedure: (**a**) Pinot noir pomace batch (200 g) with solvent (1 L); (**b**) filtering phenolic-enriched solvent; (**c**) centrifugation of phenolic-enriched solvent; (**d**) removal of solvent from extract.

**Figure 2 foods-11-01999-f002:**
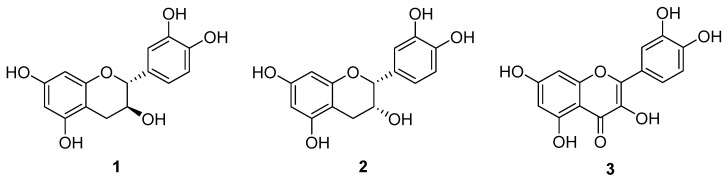
Chemical structures of the flavonoids (+)-catechin (**1**), (−)-epicatechin (**2**) and quercetin (**3**).

**Figure 3 foods-11-01999-f003:**
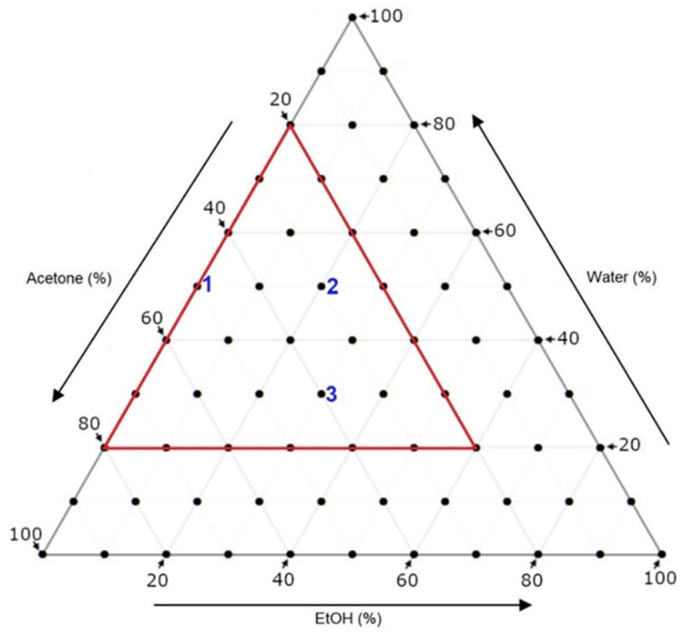
Ternary diagram with **1**, **2** and **3** (in blue) to indicate the optimal solvent systems for extracting the corresponding flavonoids from Sauvignon blanc pomace as previously identified in the literature [30] and the range of solvent systems (outlined in red) that were trialed in this study for extracting the three flavonoids from Pinot noir pomace.

**Figure 4 foods-11-01999-f004:**
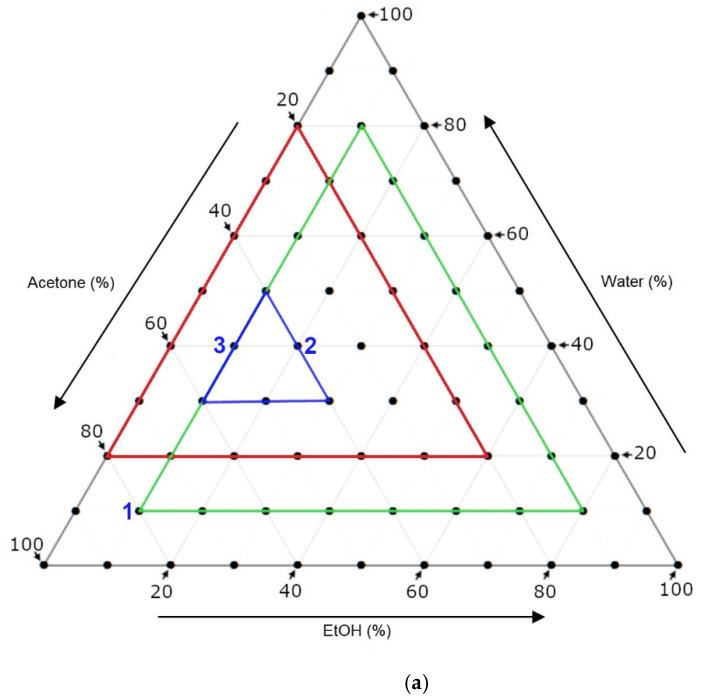
Ternary diagrams of: (**a**) the range of solvent systems trialed in this study (outlined in red), the optimal solvents for individual flavonoids (labelled 1, 2 and 3 in blue), the solvent boundaries to generate model diagrams of each flavonoid (outlined in green) and the optimal range of solvent systems for simultaneous extraction of all three flavonoids (outlined in blue); (**b**) solvent effectiveness for extraction of **1**; (**c**) solvent system effectiveness for extraction of **2**; and (**d**) solvent system effectiveness for extraction of **3**.

**Table 1 foods-11-01999-t001:** Solvent systems trialed for solvent–liquid extraction of Pinot noir pomace.

Acetone:H_2_O:EtOH						
80:20:0	60:30:10	50:30:20	40:40:20	30:60:10	30:20:50	20:50:30
70:30:0	60:20:20	50:20:30	40:30:30	30:50:20	20:80:0	20:40:40
60:40:0	50:50:0	40:60:0	40:20:40	30:40:30	20:70:10	20:30:50
60:30:10	50:40:10	40:50:10	30:70:0	30:30:40	20:60:20	20:20:60

**Table 2 foods-11-01999-t002:** Examples of flavonoid content extracted from the pomace of various grape types using SLE method; ^dw^ = dry weight.

Grape Type (Component)	Extraction Technique (Conditions)	Solvent System	Flavonoid	Average Amount (mg/kg Pomace)	Reference
Sauvignon blanc (whole pomace)	SLE (r.t., 1 h)	Acetone:H_2_O:EtOH (50:50:0)	**1**	466.29 (±83.01)	[30]
		Acetone:H_2_O:EtOH (30:50:20)	**2**	151.14 (±7.91)	
		Acetone:H_2_O:EtOH (40:30:30)	**3**	31.89 (±2.77) *	
Weisser Riesling (skins)	SLE (r.t., 2 h)	MeOH:HCl (99.9:0.1 *v*/*v*)	**1**	226.7 (±24.6) ^dw^	[45]
			**2**	134.6 (±12.1) ^dw^	
Weisser Riesling (seeds)			**1**	790.2 (±11.2) ^dw^	
			**2**	674.5 (±24.9) ^dw^	
Pinot noir (seed)	SLE (25 °C, 19 h)	MeOH	**1**	1583 ^dw^	[46]
			**3**	1386 ^dw^	
		EtOH	**1**	1450 ^dw^	
			**3**	1386 ^dw^	

* measured as rutin equivalence per kg pomace.

**Table 3 foods-11-01999-t003:** Solvent systems that had extracted the highest amount of each flavonoid and the average (of triplicate extractions) quantity (mg/kg pomace) as measured in the extract.

Flavonoid	Best Solvent System (Acetone:H_2_O:EtOH)	Average Quantity Extracted (mg/kg Pomace)
**1**	80:20:0	174.1 (±17.1)
**2**	40:40:20	269.6 (±34.1)
**3**	40:50:10	87.0 (±46.4) *

* measured as mg rutin hydrate equivalence per kg pomace.

**Table 4 foods-11-01999-t004:** The quantity of flavonoids extracted and their rank (out of 28) for each of the six solvent systems identified as optimal systems for the SLE for large-scale extraction.

Acetone:H_2_O:EtOH	(+)-Catechin (1)		(−)-Epicatechin (2)		Quercetin (3)	
	mg/kg Pomace	Rank	mg/kg Pomace	Rank	mg/kg Pomace	Rank
60:30:10	148.4 (±25.7)	16	218.8 (±39.5)	16	49.0 (±14.3)	16
50:40:10	157.0 (±32.1)	10	231.1 (±39)	12	66.5 (±4.2)	4
50:30:20	165.9 (±16.3)	7	250.86 (±25.3)	5	72.3 (±39.1)	2
40:50:10	169.9 (±28.3)	3	227.1 (±18.8)	14	87.0 (±46.4)	1
40:40:20	173.3 (±23.9)	2	269.59 (±34.1)	1	55.0 (±1.3)	9
40:30:30	164.7 (±18.4)	8	240.1 (±8.9)	9	58.9 (±01.0)	7

**Table 5 foods-11-01999-t005:** Quantities of flavonoids extracted from small-scale (200 g) and large-scale (2 kg) Pinot noir pomace using 40:30:30 (acetone:H_2_O:EtOH) solvent system.

Flavonoid	Extraction on Small-Scale Pomace (mg/kg Pomace)	Extraction on Large-Scale Pomace (mg/kg Pomace)
**1**	164.7 (±18.4)	14.0 (±5.3)
**2**	240.1 (±8.9)	1.1 (±0.6)
**3**	54.9 (±1.0) *	29.7 (±1.1) *

* measured as mg rutin hydrate equivalence per kg of pomace.

**Table 6 foods-11-01999-t006:** Cell proliferation rates of cell lines treated with non-derivatized and derivatized extracts.

Crude Extract Type	% Cell Proliferation (Compared to Control)	
	HCT116	MDA-MB-231
Non-derivatized	95.6 (±2.2)	75.1 (±27.8)
Octanoyl	91.9 (±4.1)	82.6 (±21.0)
Lauroyl	91.4 (±4.2)	88.9 (±15.0)
Palmitoyl	88.6 (±6.6)	75.8 (±24.9)

## Data Availability

Not applicable.

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
