# Peer review of "Attempts to Create Products with Increased Health-Promoting Potential Starting with Pinot Noir Pomace: Investigations on the Process and Its Methods"

_foods, 2022, doi:10.3390/foods11141999_

Round 1

Reviewer 1 Report

Manuscript 1757154

Journal Foods

Title Investigating a process of converting products extracted from Pinot noir pomace into products with improved health-promoting potential

The paper entitled “Investigating a process of converting products extracted from Pinot noir pomace into products with improved health-promoting potential” describes the extraction by solid liquid extraction (SLE) of flavonoids from grape pomace, their derivatisation and the assessment of their antiproliferative activity using different cell lines. The topic is intriguing. However, authors did not find antiproliferative activity. All the manuscript should highlight this result and authors should pay attention to claim their derivatives “biologically active”. Moreover, antiproliferative activity results are not reported in the manuscript. A major revision is suggested. Please follow the comments in the file.

Reviewer 2 Report

The submitted article covers a process that integrated extraction and derivatization methods as a potential approach for utilizing grape pomace for health-promoting products. In addition, the article examines a possible solution for the environmental concerns of winery waste management. Therefore, the practical significance of the presented research is unquestionable. The issues introduced in the content are represented clearly, and knowledgeable. The authors should address the following:

Page 3, for correlation coefficient small r should be used.

Page 6, Table 3, rewrite solvent system formula containing water.

Page 7-8, Ternary diagrams have an unreadable small font, please make it more visible.

Page 9, line 303, please cite the previously reported work about multi-step synthetic approaches.

Page 9, line 324, provide references for the studies against colon HCT116 and breast MDA-MB-231 cancer cell lines.

Page 10, The conclusion should be rewritten without paraphrasing the whole article, it is more suitable for the abstract rather than the conclusion; please just point out the results and findings.

Page 12, Please, prepare the references according to the instructions for the authors.

Round 2

Reviewer 1 Report

Authors significantly improved the manuscript following reviewer's suggestions. Minor comments are reported below:

L1066-1068 Rewrite. It is not clear

L1068-1102 Please short this part to one third. Thanks

L1158-1164 Please better describe this point. It is not clear

L1180-1182 Please better describe the selection of this solvent system. It is very important

Section 3.4 and Table 6 Please discuss the data on antiproliferative activity of derivatives. Are there other papers on the antiproliferative activity of flavonoids derivatives? 

Table 6 Please add a statistical analysis. 88.6 could be statistically different from 95.6.
